

# A spatio-temporal reconstruction of sea-surface temperatures in the North Atlantic during Dansgaard-Oeschger events 5-8

Mari F. Jensen[1], Aleksi Nummelin[2], Søren B. Nielsen[3], Henrik Sadatzki[1], Evangeline Sessford[1], Bjørg Risebrobakken[4], Carin Andersson[4], Antje Voelker[5], William H. G. Roberts[6], and Andreas Born[7,1]

[1]Department of Earth Science, University of Bergen and Bjerknes Centre for Climate Research, Bergen, Norway
[2]Geophysical Institute, University of Bergen and Bjerknes Centre for Climate Research, Bergen, Norway and Department of Earth and Planetary Sciences, Johns Hopkins University, Baltimore, USA
[3]Climate and Geophysics, Niels Bohr Institute, University of Copenhagen, Copenhagen, Denmark
[4]Uni Research Climate, Bjerknes Centre for Climate Research, Bergen, Norway
[5]Instituto Português do Mar e da Atmosfera, Lisbon, and Centre of Marine Sciences (CCMAR), University of Algarve, Faro, Portugal
[6]University of Bristol, Bristol, England
[7]Institute of Physics and Oeschger Centre for Climate Research, University of Bern, Bern, Switzerland

*Correspondence to:* M. F. Jensen (mari.f.jensen@uib.no)

**Abstract.** Here we establish a spatio-temporal evolution of the sea-surface temperatures in the North Atlantic over Dansgaard Oeschger (DO) events 5-8 (c. 30-40 ka) using the proxy surrogate reconstruction method. Proxy data suggest a large variability in North Atlantic sea-surface temperatures during the DO-events of the last glacial period. However, proxy data availability is limited and cannot provide a full spatial picture of the oceanic changes. Therefore, we combine fully coupled, general circulation model simulations with planktic foraminifera based sea-surface temperature reconstructions to obtain a broader spatial picture of the ocean state during DO-events 5-8. The resulting spatial sea-surface temperature patterns agree over a number of different general circulation models and simulations. We find that sea-surface temperature variability over the DO-events is characterized by colder conditions in the subpolar North Atlantic during stadials than during interstadials, and the variability is linked to changes in the Atlantic Meridional Overturning circulation, and in the sea-ice cover. Forced simulations are needed to capture the strength of the temperature variability and to reconstruct the variability in other climatic records not directly linked to the sea-surface temperature reconstructions. Our results are robust to uncertainties in the age models of the proxy data, the number of available temperature reconstructions, and over a range of climate models.

# 1 Introduction

The Dansgaard-Oeschger (DO) events of the last glacial are some of the most prominent climate variations known from the past. Ice cores from Greenland show multiple temperature excursions during the last glacial period as the climate over Greenland alternated between cold stadial (Greenland Stadial, GS), and warmer interstadial (Greenland Interstadial, GI) conditions with



a period of roughly 1500 years (Grootes and Stuiver, 1997). Each DO-event is characterised by an initial temperature rise of $10\pm5°$C toward GI conditions in a few decades, a more gradual cooling over the following several hundreds of years, and a relatively rapid temperature drop back to GS at the end of most of the events (Johnsen et al., 1992; Dansgaard et al., 1993; North-Greenland-Ice-Core-project members, 2004; Kindler et al., 2014). DO-events are manifested not only in Greenland,

but around the world. Concurrent with warm GIs, proxies show a warmer and wetter climate in Europe, an intensified Asian summer monsoon, and a cooling of parts of the Southern Hemisphere (Voelker, 2002; Rahmstorf, 2002, for overviews).

With its proximity to Greenland, much attention has been paid to the North Atlantic in reconstructing the millennial-scale GS/GI climate cycles. Reconstructions from Marine Isotope Stage 3 (MIS3, 24-59 ka), during which about 15 DO-events occurred, suggest a large variability in sea-surface and sub-surface temperatures in the North Atlantic coincident with the

temperature variability on Greenland. Elliot et al. (2002) and Bond et al. (1993) show significant decreases in sea-surface temperatures (SST) in the North Atlantic during GSs. In the Nordic Seas, on the other hand, sub-surface temperatures are found to increase during the same periods (Rasmussen and Thomsen, 2004; Dokken et al., 2013; Ezat et al., 2014). Large movements of the oceanic temperature fronts are associated with the variability (Voelker and de Abreu, 2013; Eynaud et al., 2009) and Rasmussen et al. (2016) recently suggested a gradual northward movement of warm subsurface waters during GSs compared

to GIs in the North Atlantic. New studies also suggest variability in the sea-ice cover of the Nordic Seas with expanded sea-ice cover during GSs compared to GIs when the sea-ice cover retreated. These changes coincide with the temperature variability on Greenland (Dokken et al., 2013; Hoff et al., 2016). However, expanded sea-ice cover during GIs compared to GSs has also been suggested based on dinoflagellate cyst assemblages (Eynaud et al., 2002; Wary et al., 2016).

Several mechanisms involving the North Atlantic have been proposed to explain the GS/GI cycles. These include latitudinal

shifts in the North Atlantic Deep Water formation site (Labeyrie et al., 1995; Ganopolski and Rahmstorf, 2001; Arzel et al., 2010; Colin de Verdiere and Raa, 2010; Curry et al., 2013; Sevellec and Fedorov, 2015); changes in the heat transport to the North Atlantic due to either internal instabilities in the Atlantic Meridional Overturning circulation (AMOC, Broecker et al., 1990; Tziperman, 1997; Marotzke, 2000; Ganopolski and Rahmstorf, 2001) or a salt oscillator (Peltier and Vettoretti, 2014; Vettoretti and Peltier, 2016); changes in the sea-ice cover of the Nordic Seas (Broecker, 2000; Gildor and Tziperman, 2003;

Masson-Delmotte et al., 2005; Li et al., 2005; Dokken et al., 2013; Petersen et al., 2013). However, the mechanisms behind the DO-events are still debated, and it is not clear whether the events are forced by internal ice-sheet instabilities, if they originate from variability within the ocean-atmosphere system, or from a combination of both.

To understand the dynamics behind the DO-events, integrating climate modelling and paleo-reconstructions is necessary. While modelling studies can demonstrate feasibility of all of the above mentioned mechanisms for DO variability, proxy data

should be used to assess the realism of such scenarios. However, due to limited proxy data and computing resources, model-data integration poses a challenge to the community. Although the North Atlantic region contains much information from the last glacial period, marine paleoclimatic reconstructions are still confined to suitable locations for coring, making the information sparse and spatially limited. In addition, the marine proxy records that cover MIS3 have variable temporal resolution, and even the best resolved marine reconstructions are of much lower resolution than the ice core records from Greenland. Although there

are a number of idealized attempts to simulate the full MIS3 period (e.g., Brandefelt et al., 2011; Van Meerbeeck et al., 2009;





Peltier and Vettoretti, 2014) the lack of high resolution coupled climate model simulations (including interactive ice-sheets) is arguably another limiting factor for our understanding of the mechanisms behind DO-events.

The model-data integration for the recent past has been performed using various techniques. One example is regression models that base on the relationship between climate model variables or observations, and are applied (directly) to proxy reconstructions (e.g., Rahmstorf et al., 2015). While using regression models is computationally efficient, it inherently assumes the same linear relationships between variables in the past as for the present; an assumption which might not be true, especially when investigating a glacial climate. Regression models have also been applied using climate models with MIS3 boundary conditions (Zhang et al., 2015), however, linearity between the model variables is still assumed and this assumption may be invalid during the abrupt non-linear DO-events. Another example is data-assimilation and inversion techniques which are becoming more frequently used for paleoclimate studies. For example, Kurahashi-Nakamura et al. (2014) and Gebbie et al. (2015) have used these techniques to investigate the ocean circulation during the last glacial maximum (LGM). However, due to the lack of proxy data, these studies often focus on reconstructing a relatively stable climate state, which allows for aggregating a large number of proxy records over a long timespan. While data-assimilation would be an ideal method for confining model simulations, it is not feasible in the transient case of MIS3 where the proxy data coverage is very sparse. Due to their individual restrictions, both regression methods as well as data-assimilation, together with long coupled climate model simulations, appear sub-optimal for studying the non-linear changes over the long time period of MIS3.

In this study, we combine the physically consistent output of model simulations with the temporally accurate information from proxy data in the North Atlantic by applying the proxy surrogate reconstruction (PSR) method. This method has recently undergone sensitivity tests and has been successfully applied in reconstructing European atmospheric temperatures over the last 200 years (Franke et al., 2011), and the observed global decadal temperature variability of the last century (Gómez-Navarro et al., 2017). Although Graham et al. (2007) used one coastal SST reconstruction together with terrestrial records to reconstruct US climate back to 500 AD, the PSR method has, to our knowledge, never been applied to ocean data before. Also, the PSR method has never before been used for the MIS3 time period. Here, we use the SST variability in the North Atlantic during the last glacial period as a test-case for the PSR method to see whether the method can widen our knowledge of spatial patterns back in time. In doing so, we also aim to learn about the underlying DO variability in the North Atlantic region.

Our analysis shows that the surrogate reconstruction is a good solution for combining model simulations and proxy data without having to assume the same climatic changes at all locations. In the North Atlantic, where the data coverage during MIS3 severely limits the usefulness of other data-assimilation and inversion techniques, the PSR method is an especially useful method. As the method allows for the full non-linearity of the climate system, as represented by the coupled model simulations, we are able to capture a large part of the glacial variability in the ocean during MIS3 and also on Greenland. The method is an efficient offline data assimilation technique which allows for the quantification of uncertainty, unlike expensive transient simulations, which makes it a good first step for understanding the spatial variability of the climate changes seen during MIS3. Until such time as full complexity models are capable of simulating the full transient evolution of the climate over long periods, such simplified approaches will be crucial to our understanding of the climate.





We present the PSR method together with the proxy-based temperature reconstructions and model simulations in Sect. 2, test the method in Sect. 3, show the results of the method in Sect. 4, and discuss the results in Sect. 5.

## 2  Materials and Methods

### 2.1  Proxy Surrogate Reconstruction

We use the PSR method which was first introduced by Graham et al. (2007) to combine sparse proxy data and spatially complete and physically consistent climate model output. In this way we can produce a climate reconstruction that is complete in both space and time. This approach was first introduced for weather forecasting by Lorenz (1969).

Briefly, the PSR method is as follows (see below for details). A collection of years from one or several model simulations are treated as a pool of possible "climate states" (see Sect. 2.3 for details). Then, for any time period in which we have a set of
proxy data, we find the climate state from the model pool that best matches the proxy data: an "analog". This is done using an objective cost function. By repeating this and finding analogs for all time steps in the proxy data, we compile a set of climate states from the model pool which are consistent with the proxy data. We thus have a time series of modelled climate states with a complete spatial coverage which are the best fit to the spatially sparse proxy data. Furthermore, since the model simulates variables other than those that are used in the matching, we can also reconstruct variables for which no paleo-reconstruction
exist.

We follow the PSR implementation introduced by Franke et al. (2011). First, we find the model grid cells that are closest to the proxy record locations and extract the modelled temperature for each climate state from those locations. Results are not sensitive to this choice as picking the four closest grid cells yield virtually identical results. Second, we define a cost function for deciding which of the climate states will represent a given proxy time step. For this we use the root mean square error
(RMSE) in temperature space as a distance measure:

$$d(T^p, T^m) = \sqrt{\frac{\sum_{i=1}^{I} w_i (T_i^p - T_i^m)^2}{I}}, \tag{1}$$

where $T_i^p$ and $T_i^m$ are the proxy and model temperature anomaly records at location $i$, respectively. The anomalies are calculated from the temporal mean of the proxy data and each individual model simulation, respectively (see. Sect. 3.2.3). $I$ is the total number of core locations with a proxy-based reconstructed temperature value at the given time step, ranging from 12-14
in this study. $w_i$ is a weight which one could apply to reduce the bias toward areas with large variability. However, we decided not to use any weight ($w_i = 1$), because the spatial differences in interannual variability in the ocean are much smaller than spatial differences in monthly variability in the atmosphere (which is what motivated Franke et al. (2011) to use $w_i$). We did test several different weighting options, however, no weighting function impacted the final results, further justifying our choice of setting $w_i = 1$.

As a final step, we create a composite $T^c$ consisting of several analogs; the model climate states with the smallest $d$. We choose to form the composite based on the 10 analogs (N=10) with the smallest RMSE, and weight the analogs by a function



of the RMSE;

$$T^c(x, y, t = k) = \sum_{n=1}^{N} W_n T_n^m(x, y) \tag{2}$$

where $x$ and $y$ are the two spatial dimensions, $t$ is time dimension (on the proxy time axis), $k$ marks a given year in the proxy record, $n$ is a specific analog from the model pool, $T_n^m$ is the model temperature field for analog n, and the weight scalar $W_n$ is defined to be inversely proportional to the RMSE distance $d_n$:

$$W_n = \frac{\left(\sum_{n=1}^{N} d_n\right) - d_n}{\sum_{n=1}^{N}\left[\left(\sum_{n=1}^{N} d_n\right) - d_n\right]}. \tag{3}$$

Note that because the sum of $W$ is scaled to be 1, the composite is not very sensitive to the increase in number of analogs ($N$) as usually analogs beyond $n = 10$ are assigned with very small weights. The choice of number of analogs in the composite is further discussed in Sect. 3.2.1.

As we repeat this algorithm for each proxy time step, we construct a three-dimensional dataset of the composites; the surrogate reconstruction $c_i$. We either present the results as a surrogate time series or as composite maps of the surrogate reconstruction.

## 2.2 Data Pool

We compiled proxy-based SST reconstructions from 14 marine sediment cores located in the North Atlantic for the time interval from 30 to 40 kyr (GS/GI cycle 5-8) (Table 1, Fig. 1). For 10 of the cores (Table 1), we calculate SST estimates based on assemblage counts of planktic foraminifera and the Maximum Likelihood technique (ML) (ter Braak and Looman, 1986; ter Braak and Prentice, 1988; ter Braak and van Dam, 1989). We use the North Atlantic core top calibration dataset developed within the MARGO framework (Kucera et al., 2005). The calibration uses modern SST values for 10 m water depth during summer (July, August, September, JAS) taken from the World Ocean Atlas version 2 (WOA, 1998). The choice of transfer function is most often based on the RMSE between the measured and inferred variable, used to access the predictive power of the transfer function. Choosing a transfer function with low RMSE will in a statistical sense provide the best transfer function model. The estimation of the predictive power of transfer functions assumes that the test sites are independent from the modelling sites. However, autocorrelation is common in ecological data. Using an independent dataset, Telford and Birks (2005) explored and quantified how autocorrelation affects the statistical performance of different transfer functions. They concluded that the true RMSE is approximately twice the value of previously published estimates for some methods, e.g., the modern analog technique. In this study we have chosen the ML transfer function technique which is based on a unimodal species response. We base this decision on the results by Telford and Birks (2005) that suggest that this method, and other methods based on the assumption of an unimodal species-environment response model, is more robust with regards to spatial structure in the data. For the ML technique the RMSE is 1.78°C, based on cross-validation using bootstrapping.



For the remaining four cores from which the full planktic foraminifera assemblage counts are not available, we use records of the percentage of the polar planktic foraminifera *Neogloboquadrina pachyderma* (%NP, formerly known as N. pachyderma sinistral) to reconstruct the SST variations. We include these four additional records to increase the spatial coverage of SST reconstructions. SST estimates based on %NP rely on a linear relationship between SST at 10 m water depth and %NP as

described by Govin et al. (2012), where

$$\text{SST} = -0.06\%\text{NP} + 12.26. \tag{4}$$

As the linear relationship in Eq. (4) is only valid for %NP values between 10 and 94%, we exclude values out of this range. Cores 5 and 9 have all values within the interval, while for cores 6 and 10 we have excluded parts of the record.

We use the original age model of each core and linearly interpolate between existing data to obtain data points at consistent

steps of 20 years. The absolute temperatures are shown in Fig. 2, but note that we use the temperature anomalies from the temporal mean over the 30-40 kyr interval in the PSR method. The age models of all but three cores are defined by correlating SST or Ice Rafted Debris (IRD) signals to the DO signals in the GISP2 ice core record, or magnetic susceptibility signals to those in the NGRIP record (Table 1). In the Greenland ice cores the transitions between stadials and interstadials occur in less than 50 years (Rasmussen et al., 2014). The difference between the original GISP2 and NGRIP chronologies is up to 300

years in between 40 and 30 kyr (Svensson et al., 2008). The transitions between the stadials and interstadials are rapid, large scale features that can easily be identified. However, when the respective marine property records are tuned to the Greenland isotopes, an added uncertainty to the relative relation between the dated records is expected. Taking into account the various sources of age model uncertainties, we argue that a ±500 years relative age uncertainty in between the records that have been tuned to Greenland ice core records is a conservative estimate. As the PSR method is sensitive to relative dating differences,

we test how our results are influenced when assuming a ±500 year age uncertainty. We note that the original age models of cores 3, 4 and 14 were solely based on [14]C dates (Table 1), and the age uncertainty for these cores, relative to the ages of the other cores, is considered larger.

## 2.3 Model Pool

We base the surrogate reconstruction on a number of different GCM simulations. The main model pool consists of model output

from the Hadley Centre coupled model version 3 (HadCM3, Gordon et al., 2000; Singarayer and Valdes, 2010), spanning 8 simulations at different times. There are 6 time slices from MIS3; at 30, 32, 34, 36, 38, and 40 kyr. In addition, a time slice at 21 kyr (LGM), and a pre-industrial (PI) simulation are included. Each simulation is initialised from a PI simulation and run for 500 years, and the last 200 years are included in the pool. The simulations are performed with prescribed orbital forcing (Berger and Loutre, 1991), ice sheet configuration and greenhouse gases (Petit et al., 1999; Loulergue et al., 2008; Spahni

et al., 2005) corresponding to the respective ages. Since there are no ice sheet reconstructions for MIS3, which is before the LGM, we use ice sheets that are a linear rescaling of their LGM topography. To do this rescaling, we multiply the LGM ice sheet height by the fraction of the total LGM ice sheet volume that is realised at each time slice. We use the ICE-5G LGM ice sheet reconstruction (Peltier, 2004) and assume that for all MIS3 time slices the ice sheet extent is as for the LGM: it is




only the ice sheet height that varies. See (Singarayer and Valdes, 2010) for further details. Additionally, we add 99 years from two simulations (32 and 38 kyr) where 1 Sv of freshwater is added to the Atlantic between 50°N and 70°N to mimic Heinrich events (for more details, see Singarayer and Valdes, 2010). For this study, we continue these two simulations for 250 years with the freshwater forcing turned off. The horizontal grid of the ocean is $1.25° \times 1.25°$. These 10 simulations constitute the

main model pool, but we also perform sensitivity experiments with the PSR method using the 8 unforced simulations only ("HadCM3 nohose").

Additionally, we experiment with the Community Climate System Model version 4 (CCSM4) data from a 1300 year long PI simulation and a 200 year long LGM (only 1000 and 100 years, respectively, included in the model pool) simulation of the $1° \times 1°$ version (Gent et al., 2011; Danabasoglu et al., 2012; Brady et al., 2013) as well as a 1000 year long PI simulation with a $2° \times$

$2°$ atmosphere model (Kleppin et al., 2015; Born and Stocker, 2014). Including the $2°$ atmosphere version of CCSM4 adds new model states to the pool as the simulation has been shown to have cold GS-like, and warm GI-like conditions (Kleppin et al., 2015).

As the proxy data are calibrated to 10 m depth and represent a summer temperature average, the model pools consist of temperatures from the upper vertical grid cell averaged over the months JAS. As the proxy data in essence represents an

integrated signal of temperatures over several years, the JAS averages from the model output are low-pass filtered using a 4th order Butterworth filter with a cut off frequency of $0.2 \, \mathrm{yrs}^{-1}$, thus each member of the model pool is a 10 year average of model data. For all GCM simulations, we remove the temporal mean of each simulation to obtain temperature anomalies for the PSR method. This is also done for all other variables when using anomalies. Different ways of defining the anomalies are discussed in Sect. 3.2.3.

The model pools are summarized in Table 2 where the main model pool, HadCM3, is highlighted.

## 3    Testing the PSR method

Here, we test the PSR method, first with synthetic data (Sect. 3.1), and then with the proxy data and model pools (Sect. 3.2) introduced in Sects. 2.2 and 2.3.

### 3.1    Synthetic PSR-study

Before applying the PSR method on the proxy reconstructions, we perform a test with synthetic data. The synthetic data are 30 random climate states (as explained above) of the HadCM3 model output at the proxy locations. The remaining climate states from the model output serve as the model data pool. We follow the PSR method as described in Sect. 2.1 with the final surrogate reconstruction based on an average of 10 analogs. We perform such a reconstruction 1000 times in order to obtain robust statistics, and find the distance between the PSR output and the original model output at each grid cell. This is presented

as offsets in Fig. 3 at the 5, 50, and 95 percentile.

The good agreement between the PSR output and the original model output (Fig. 3) demonstrates that the spatially sparse synthetic data at the proxy locations can effectively recover the simulated ocean state. The correlation between the surrogate





reconstruction and the original model output exceeds 0.8 in the North Atlantic, except along the eastern coast of Greenland and along 35°N where the correlation still exceeds 0.6. Practically all of the variability is captured in the eastern subpolar gyre. The reconstructed temperatures are generally within $\pm 0.1$°C of the model output (Fig. 3b), but some of the reconstructed values are up to 1°C from the real value. These offsets are mainly found in the Nordic Seas region and along the coast of North

America (Fig. 3c,d). This reflects the fact that the proxy locations do not provide enough information for the PSR method to fully recover the model output in these areas. Using the CCSM4-model pool as the observations and the HadCM3 as the model data pool, the largest offsets are 1.5°C (not shown). In general, these results act as a guide as we proceed to the proxy data.

## 3.2 Proxy data testing

### 3.2.1 Number of analogs

Several climate states from the model pools might match the proxy-reconstructions well at the core locations, but differences away from the core sites might be large as different climate states can produce similar RMSE at the core locations (Fig. 4). Therefore, to ensure a consistent surrogate reconstruction, the composites are an average of a number of analogs with the smallest RMSE. As the number of analogs (N) increases, each additional member has less and less effect on the final 2D pattern. As noted by Gómez-Navarro et al. (2017), the correlation between the proxy and the surrogate time series increases

with N, while the standard deviation of the surrogate time series decreases with it. In our case, choosing N much larger than 10 results only in a small increase in correlation, but in a large decrease in the standard deviation of the surrogate time series (Fig. 5a). On the other hand, choosing N less than 10 decreases the correlation of the time series. In addition, the lowest RMSE is obtained around N=10, the number of analogs which we use for the remainder of this study.

### 3.2.2 Model pool

The correlation between the surrogate reconstructions and the corresponding proxy data is largely independent of the model pool, while the amplitude (standard deviation) of the variability depends on it (Fig. 5b). The surrogate reconstruction based on the full HadCM3 model pool outperforms the others with regard to the standard deviation and RMSE (Fig. 5b), which is why it is the main model pool in this study. The mean correlation between the proxy record and the surrogate time series at the core locations does not change when excluding the two hosing simulations from the model pool. However, the mean standard

deviation of the surrogate time series at the core locations decreases from 0.98 with the main HadCM3 model pool to 0.64 with HadCM3 nohose. If the CCSM4 pool is used instead, the variability of the surrogate time series decreases further.

For all model pools, the JAS average of the model data gives better results than annual means. As expected, unfiltered data give a larger variability of the surrogate time series than filtered data. However, as the proxy data consist of temperature signals integrated over several years, only filtered data are used hereafter.



### 3.2.3 Model-proxy distance

Ideally, if the full model pool captures the variability of the proxy record, the best analogs are scattered among all possible climate states represented in the model pool and have equally low RMSE (Eq. (1)) throughout the surrogate reconstruction. This is not the case throughout the record (Fig. 4) as generally colder periods are better captured by the model pool than warmer

periods. The worst fit between the model and proxy data is found during the time interval 36.9-38.3 kyr (same for all models; not shown). During this interval the RMSE is large and the best analogs are constructed from very few different climate states which mainly belong to the hosing simulations. One could argue that this is an artifact that depends on the baseline from which we calculate the anomalies for both proxy and model data. Indeed, the mean of the proxy data is closest to stadial temperatures and the amplitudes of the warm interstadials are larger than the cold stadials in the proxy data. Therefore a large RMSE during

interstadials does not necessarily indicate that the model state is furthest from the interstadials, but rather that the model pool does not capture the full variability of the proxy record.

Nevertheless, because choosing a baseline from which we compute the anomalies is not trivial, we test three plausible alternatives.

For the original approach, we simply compute anomalies from the temporal mean;

$$
\begin{aligned}
T_i^{p'} &= T_i^p - \overline{T_i^p}, \\
T_i^{m'} &= T_i^m - \overline{T_i^m},
\end{aligned}
\tag{5}
$$

where the overline ($\bar{\cdot}$) represents the mean, and the prime ($'$) the anomaly (excluded in the other sections of the text for simplicity). Note that for the model pool the anomalies are with respect to each individual model simulation, not the full model pool.

One could argue that a reasonable way to calculate the proxy and model anomalies would be to compare them to PI

(Holocene) conditions in order to avoid the bias toward stadial times. We have temperature reconstructions of PI conditions for 9 of the cores for which we use the mean of the past 10 kyr as the baseline. We omit the proxy records for which we do not have this information. For the model pools we use the mean of the PI control simulations. The anomalies become:

$$
\begin{aligned}
T_i^{p'} &= T_i^p - \overline{T_i^p}(0 - 10\,\mathrm{kyr}), \\
T_i^{m'} &= T_i^m - \overline{T_i^m}(\mathrm{PI}).
\end{aligned}
\tag{6}
$$

Using the PSR method with these anomalies we find that the RMSE between the proxy data and the model simulations

increases, and the mean correlation between the surrogate and proxy reconstructions at the core locations drops by 35% (compared to the original definition, but with only 9 cores), and very few different model years are chosen for the analogs.

Finally, one could argue that a reasonable choice would be to use the stadial conditions as the baseline. Here we compute the anomalies as;

$$
\begin{aligned}
T_i^{p'} &= T_i^p - \overline{T_i^p}(38.2 - 39.9\,\mathrm{kyr}), \\
T_i^{m'} &= T_i^m - \overline{T_i^m}(\mathrm{MIS\,3\,no\,hose\,sim.}).
\end{aligned}
\tag{7}
$$



As a result, the RMSE increases, and the mean correlation and standard deviation of the resulting surrogate reconstruction and proxy data at the core locations decrease. Neither the hosing simulations, nor the PI or LGM simulations are chosen for the analogs. This is also true if Eq. (7) is normalized by the standard deviation of the stadial time period in addition. The definition of the anomaly is clearly important for the PSR method, but since the original definition performs the best, the following
discussion will focus on it.

## 4   The proxy surrogate reconstruction

We now look at the results of the PSR both as time series at the core locations and as composite maps. Our interpretation of the results is found in Sect. 5.

### 4.1   Reconstruction at proxy locations

The performance of the surrogate time series at the different proxy locations differs between sites (Fig. 6, $r$=0.47-0.92; A=0.32-1.24, where A is the ratio between the standard deviation of the surrogate and proxy data time series). The surrogate time series matches the proxy time series best in the central North Atlantic and in the subpolar gyre (cores 5,6,8,10,12, and 14; $r^2$=0.64-0.92; A=0.52-1.01). Surrogate time series taking into account possible uncertainties in the age models produce a confidence interval that generally follows the proxy time series. This is especially true for the long lasting GS9 and GI8 where most of
the surrogate time series produced with the original age models agree with the age model uncertainties at the core locations. We therefore focus on GS9 and GI8 when studying the spatial patterns using composite maps of the surrogate time series over these periods.

The surrogate reconstruction proves to be relatively independent of a single proxy time series. By excluding one core at a time before performing the PSR method, we can estimate the importance of each core in the full reconstruction. The mean correlation
between the proxy data and the surrogate time series hardly changes and slightly improves in some cases ($\overline{r} = 0.71 - 0.78$) as there are fewer constraints on the surrogate reconstruction. By doing this, we can also assess the agreement between the proxy data and the surrogate time series at the location where the core was excluded (i.e., bootstrapping). Once again, the performance differs between sites, but the agreement between the two records is on average $r = 0.35$.

### 4.2   Reconstructed spatial patterns

The surrogate reconstruction based on the full HadCM3 model pool shows colder SSTs during GS9 than during GI8 throughout the subpolar North Atlantic (Fig. 7). Notable exceptions are the subtropical-subpolar gyre boundary off the coast of North America which shows warmer conditions during GS9 than during GI8, and the eastern subpolar gyre and the northern Nordic Seas which show little to no temperature change. We note that the synthetic test in Sect. 3.1 showed that the variability in the subtropical-subpolar gyre boundary off the coast of North America was poorly represented by the variability at the core
locations. During GI8, the subpolar gyre is warmer than normal with especially strong warming near the Greenland-Scotland



ridge and the coast of western Europe. Except for the magnitude of the warming in the eastern North Atlantic, the results are robust to differences in age models and do not depend on single cores (see details in Fig. 7).

The spatial SST patterns are largely consistent with the surrogate reconstructions produced using the other model pools. Both the CCSM4 pool and the HadCM3 nohose-model pool produce comparable surrogate time series, albeit lacking the

amplitude present in the main HadCM3 model pool. The temperature patterns are similar (Figs. 8, 9), although the Nordic Seas are warmer during GS9 than GI8, which is not clearly seen when the HadCM3 model pool is used. We note that the general SST pattern holds when PI control simulations with NorESM and IPSL are used as the model pools (not shown). The similarity in reconstructed ocean patterns suggests that the ocean variability is based on physical mechanisms present in a wide range of simulations.

**4.3  Extending the information to other climate variables**

Given the reasonable agreement with the reconstructed SSTs and the original proxy record, we expand our analysis to other climate variables. We note that none of these variables are constrained by the PSR method, other than by being linked to the changing SSTs over the North Atlantic. Using the HadCM3 model pool, the ice core temperature record from NGRIP (Kindler et al., 2014) and the reconstructed atmospheric temperature from the closest grid location to NGRIP in the surrogate time series

are highly correlated (Fig. 7d, Table 3). Despite the high correlation, the amplitude of the temperature variability is lower in the PSR record than in the ice core. However, since we are not able to capture the full amplitude of the temperature variability in the ocean, we do not expect to capture the full temperature variability on Greenland either.

While the HadCM3 model pool produces reasonable agreement with ice core record at NGRIP, less of the NGRIP temperature variability is captured with the HadCM3 nohose and CCSM4 pools (Figs. 8d, 9d, Table 3). The results of NGRIP

are comparable if the surrogate reconstructions are compared to temperature reconstructions from the GISP2 ice core instead (Table 3, Cuffey and Clow, 1997; Alley, 2004).

The agreement between the reconstructed temperatures from ice cores on Greenland and the surrogate reconstruction shows that the PSR method has skill at reconstructing the climate in areas away from the core locations and in variables that are not directly linked to the SST-proxies. Therefore, we now look at more variables and outside the core locations for GS9/GI8.

Even though we do not aim to fully understand the dynamics of the variability, other variables might elucidate the mechanisms behind the oceanic variability we see.

We start by examining the ocean circulation with the AMOC variability of the surrogate reconstruction based on the HadCM3 model pool (Fig. 10), in which AMOC mostly varies with the strong freshwater forcing. The surrogate reconstruction consists of years from different states of the hosing runs: GS9 is best represented by the hosed conditions with weak AMOC, whereas

GI8 is best represented by years just before or after the hosing. The beginning of GI8 tends to be best represented by the end of the hosing simulation as the AMOC resumes, while the end of the GI8 tends to be best represented by the beginning of the hosing simulations when the AMOC weakens (Fig. 4).

The overturning streamfunction output was not available for all of the HadCM3 simulations, but we were able to continue the simulations lacking the output for another 100 years to recover the streamfunction output (note that these are simulations with



constant forcing and we do not expect the AMOC properties to change from the original data). These simulations form a new model pool HadCM3⋆ (Table 2) which enables a full AMOC reconstruction. We note that the PSR based on the HadCM3⋆ and HadCM3 are comparable (Fig. 5, similar RMSE and surrogate time series at core locations; not shown). The AMOC reconstruction shows a stronger overturning circulation during GI8 than during GS9; the whole upper cell of the streamfunction

is weaker during GS9, apart from a negative cell in the Southern Ocean which behaves the opposite way (Fig. 10d). There is hardly any change in the overturning in the Nordic Seas. The same analysis with the HadCM3 nohose and CCSM4 model pools show a generally stronger AMOC during GI8 than during GS9 (Figs. 12d, 11d). Interestingly, the HadCM3 nohose model pool shows a stronger overturning circulation over the Greenland-Scotland ridge during GI8 than GS9, but a weaker Southern Ocean cell during GS9 than GI8. However, the HadCM3 nohose and CCSM4 model pools lead to AMOC changes that are generally

very small. Due to the uncertainties in age models of the oceanic proxies, it is difficult to conclude whether the AMOC signal leads or lags the transitions between the GSs and GIs.

    Sea-ice changes are largest for the main HadCM3 pool which includes hosing. There is a ∼25% decrease in the annual sea-ice concentration in the subpolar gyre region between GS9 and GI8. The sea-ice edge retreats north-west during GI8, with largest change in the winter sea-ice edge (Fig. 10c). Close to the Greenland-Scotland ridge, the annual mean sea-ice

concentration changes are even larger (>50%) as the area is perennially ice covered during GS9, but ice-free almost the whole year during GI8. For the model pools without the hosing simulations the changes in the sea-ice cover are small (Figs. 11c, 12c) and mostly visible as a shift in the summer sea-ice edge.

    Consistent with the sea-ice changes and the freshwater forcing, the surrogate reconstruction shows a much fresher surface in the North Atlantic during GS9 than GI8 (Fig. 10b). Indeed, the surrogate time series consist of model years taken from the

beginning and end of the hosing simulations during GS9 and GI8, respectively (Fig. 4). Interestingly, the unforced simulations also produce a fresher subpolar gyre during GS9 (Figs. 11b, 12b), although the freshening does not extend to the Nordic Seas as in the freshwater forced simulations.

    Changes in the atmospheric temperature are similar to the changes in the SSTs. The HadCM3 model pool results in a colder atmosphere during GS9 than GI8 over much of the North Atlantic, with the cold conditions extending to Greenland, western

Europe and northern Africa (Fig. 10a). However, in the subtropical-subpolar gyre boundary and in the southern Atlantic, the atmosphere is warmer during GS9 than GI8, consistent with the AMOC changes. The strongest atmospheric anomalies are found near the Greenland-Scotland ridge where the sea-ice changes are the largest. When the HadCM3 nohose and CCSM4 model pools are used, the atmospheric temperature signals are mostly confined to the North Atlantic ocean and do not extend over land, except to parts of western Europe and Eastern North America (Figs. 11a, 12a). For both model pools, the subpolar

gyre region is colder during GS9 than GI8, but for the Nordic Seas and the subtropical-subpolar gyre boundary where the atmosphere is warmer during GS9 than GI8 (consistent with the SST changes).





## 5 Discussions

The PRS method is tested for the first time with proxy data from MIS3. We are able to produce a new time series which agrees with the information from the proxy data, although being dependent on the relative dating between records. We have addressed the uncertainties in the age models for the different proxy records, however, each individual test could be further evaluated.

We also note that the method gives quantitatively similar results when individual proxy records are excluded, suggesting that one poor age model would not throw off the results. Further testing could include the addition of noise to the records to see how robust the PSR is to errors in the proxy data. The method is also sensitive to how the anomalies used to compare proxy data and model output are defined, and care should be taken in finding the right analogs. One obvious limitation is the lack of long LGM and glacial simulations with different forcings, however, such simulations can easily be included in the model pool

should they come available. In general, expanding the model pool is straightforward and would be a natural next step when new simulations become available, for example through the Climate Model Intercomparison Project Phase 6 (CMIP6).

We have shown that the PSR leads to a pattern of SST change from GS9 to GI8 which is largely independent of the underlying model pool. However, the results suggest that forced simulations are needed to capture the magnitude of the SST variability and the temperature signal on Greenland. We suggest that these results have two interesting implications.

First, the pattern of SST variability can be reproduced across a wide range of model simulations including some without external forcing. This suggests that DO-events create SST patterns that are not unlike those of modern internal variability. In the case of the HadCM3 model pool, this mode is excited by freshwater forcing. However, we suggest that in the case of HadCM3 nohose and CCSM4 model pools a somewhat similar pattern results from an ocean response to a shift in the North Atlantic Oscillation (NAO), a prominent mode of atmospheric variability. This interpretation is backed up by earlier studies

that show almost identical SST (Delworth and Zeng, 2016; Delworth et al., 2017) and sea-ice (Ukita et al., 2007) patterns as a response to changing NAO, but this does not exclude the possibility for other trigger mechanisms. It is clear that the oceanic changes associated with the NAO are not large enough to explain the amplitude of the ocean variability during DO-events

Second, the PSR from the HadCM3 model pool is the only reconstruction where the GS9 to GI8 temperature difference is visible much beyond the North Atlantic Ocean. Previous literature (Broecker, 2000; Gildor and Tziperman, 2003; Masson-

Delmotte et al., 2005; Li et al., 2005, 2010; Dokken et al., 2013; Hoff et al., 2016) suggests that changes in the sea-ice cover amplify the temperature response on Greenland over the DO-transition. We suggest that the lack of sea-ice change in the HadCM3 nohose and CCSM4 model pool based PSRs is the primary reason for the weak temperature signal over Greenland in the surrogate reconstructions. Similarly, the weaker than observed temperature variability in the HadCM3 based PSR on Greenland is probably partly dependent on the amplitude and location of the changes in the sea-ice cover. Sea-ice reductions

in the Nordic Seas have been shown to produce a larger temperature response on Greenland than sea-ice reductions in the subpolar gyre (Li et al., 2010) which is where we see the largest sea-ice changes. We also note that for the HadCM3 model pool, the largest changes in the sea-ice cover take place in winter, consistent with studies suggesting that winter sea-ice change is needed to capture the Greenland temperature variability (Li et al., 2005; Denton et al., 2005). However, for HadCM3 nohose and CCSM4, the change in the summer sea ice tends to be larger than the change in the winter sea ice.





While our results suggest that changes in the freshwater input to the North Atlantic could explain large parts of the glacial temperature variability in the North Atlantic and on Greenland, this does not necessarily imply that freshwater input is the only possible forcing of the glacial variability. Since the internal variability reproduces the patterns of glacial SST variability, it could be that any forcing that can excite such internal modes of variability could possibly contribute to the DO type of temperature

change if the boundary conditions (e.g., sea ice) are right. The results suggest that even in the absence of freshwater forcing, GS9 is linked to anomalously fresh conditions in the cold subpolar gyre (Figs. 11b, 12b). Such fresh conditions provide a positive feedback that slows down the circulation and further cools the gyre, a feedback mechanism found to be important in both decadal and millennial climate variability (Born et al., 2010b; Born and Levermann, 2010a; Born et al., 2013; Moffa-Sánchez et al., 2014; Born et al., 2015). Therefore, even if the freshwater is not a primary forcing needed for glacial variability,

internal freshwater feedbacks like the subpolar-gyre response can still be an important amplifier in the glacial variability. Extending the model pool to simulations forced with other possible mechanisms could be used to further explore the importance of the freshwater forcing.

## 6   Conclusions

We have applied, for the first time, the proxy surrogate reconstruction method to oceanic proxy data from MIS3. The results

are robust to different sensitivity tests and relatively independent of the underlying model data and proxy locations.

Our results imply that the patterns of oceanic variability in the glacial climate can be well represented by patterns of short term variability intrinsic to the climate system. We find consistent patterns of variability in sea-surface temperature, sea-surface salinity, and atmospheric temperature in two different climate models and in different background climates. However, to fully capture the amplitude of the North Atlantic sea-surface temperature variability, and the surface air temperature variability on

Greenland, forced simulations, in our case forced by freshwater, are required.

Our results further suggest that the sea-ice cover could play an amplifying role during DO-events. We see a clear reduction in sea-ice concentration between stadial and interstadial conditions when the forced simulations are included in the proxy-surrogate reconstruction, likely contributing to the temperature signal on Greenland. Indeed, the lack of a sea-ice signal might be one of the reasons why the unforced model results lack the amplitude of the DO variations, especially on Greenland.

Finally, we are well aware of the limitations of the methodology and the validation so far back in time. Therefore, we see this study as an encouraging first attempt to apply the PSR method to oceanic variability during MIS3 and believe it should be further developed for an ideal implementation. A straightforward next step would be to expand the model pool with simulations from different models and with different boundary conditions.

*Data availability.* All proxy records used (with exception of the MD95-2010 record) are previously published, and we refer to the original

publications for details regarding availability of these records. The record MD95-2010 is currently under publication processes, and details



will be added when they become available. For access to the model output from HadCM3 please contact Dr. William H. G. Roberts directly (William.Roberts@bristol.ac.uk). The model output from the CCSM4 can be found at e.g., http://pcmdi9.llnl.gov/.

*Competing interests.* The authors declare that they have no conflict of interest

*Acknowledgements.* The research leading to these results is part of the ice2ice project funded by the European Research Council under the
5   European Community's Seventh Framework Programme (FP7/2007-2013)/ERC Grant Agreement No. 610055. The research was supported by the Centre for Climate Dynamics at the Bjerknes Centre. We thank Paul Valdes and Joy Singarayer for the use of their HadCM3 model simulations. The HadCM3 model simulations were carried out using the computational facilities of the Advanced Computing Research Centre, University of Bristol - http://www.bris.ac.uk/acrc/. We acknowledge the World Climate Research Programme's Working Group on Coupled Modelling, which is responsible for CMIP, and we thank the National Center for Atmospheric Research for producing and making
10  available their model output.



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

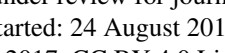
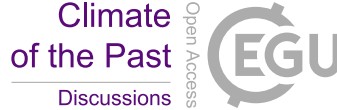

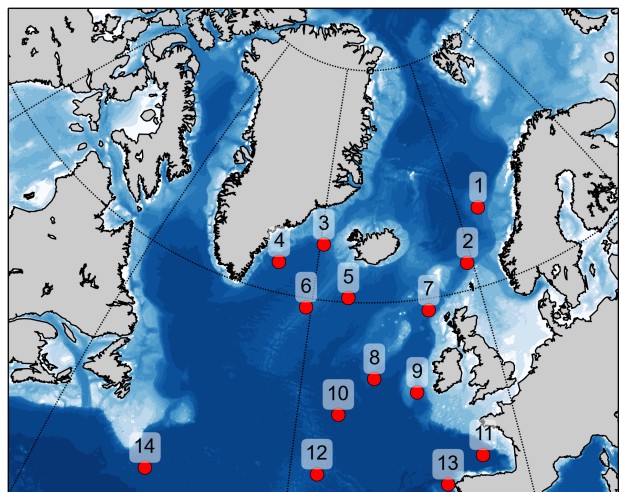

**Figure 1.** Core locations of the proxy data, numbers correspond to Table 1. Colors indicate bathymetry.




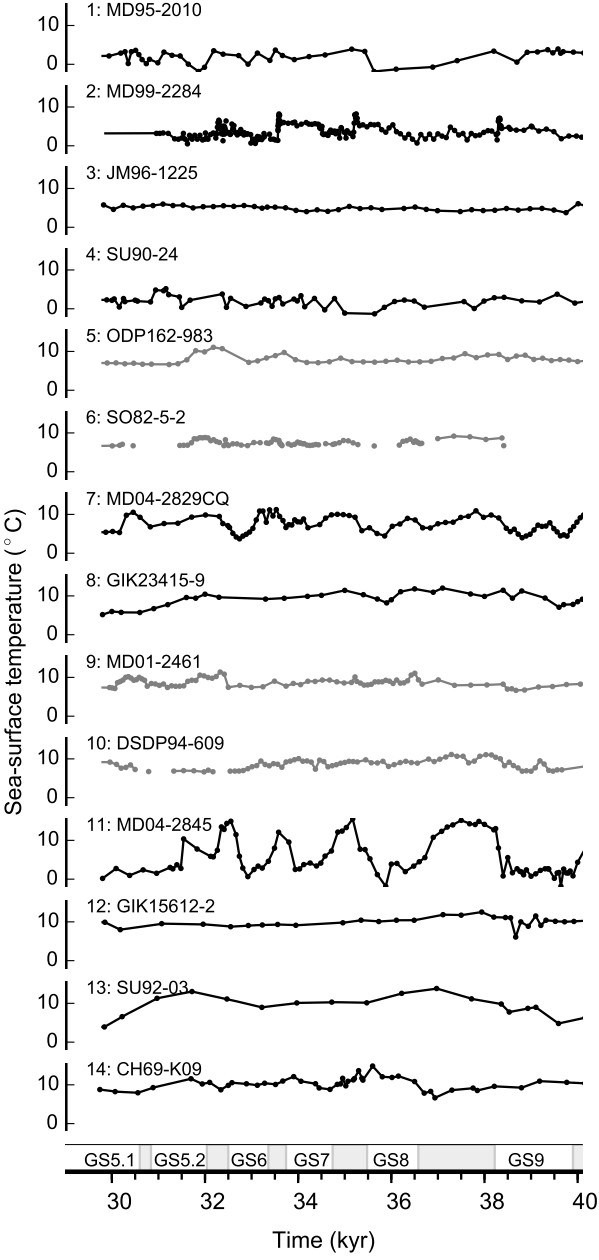

**Figure 2.** The proxy data. Black time series correspond to ML SSTs while grey time series are produced with %NP. Dots mark a data point. Shaded, grey areas on x-axis mark the GIs, while the GSs are named.





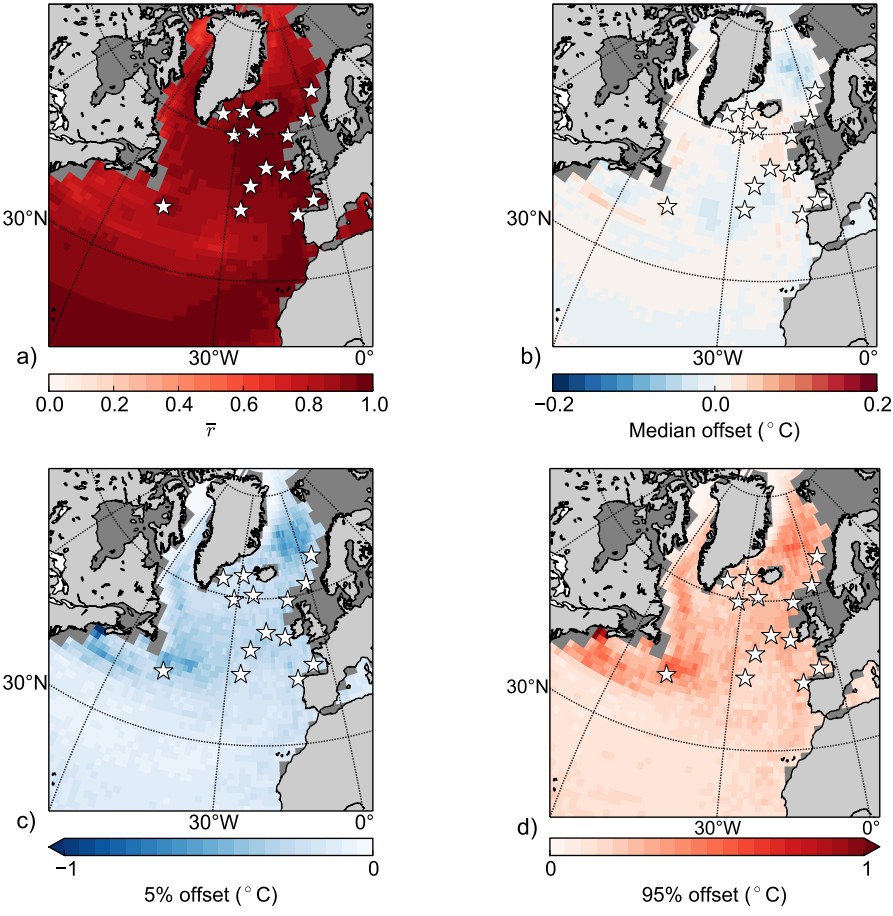

**Figure 3.** Synthetic PSR-study. Colors show the a) mean correlation coefficient between the synthetic data and the surrogate reconstruction, b) the median, c) the 5% and d) 95% offset between the synthetic value and the reconstructed surrogate value. The synthetic observations are 30 random years from the main HadCM3 model pool at the core locations, the model pool is the HadCM3, synthetic data extracted. This is repeated 1000 times. Grey shading represents the glacial coastline in the tcmoe3 simulation (Table 2, differs slightly between runs), while the white stars mark the core locations.





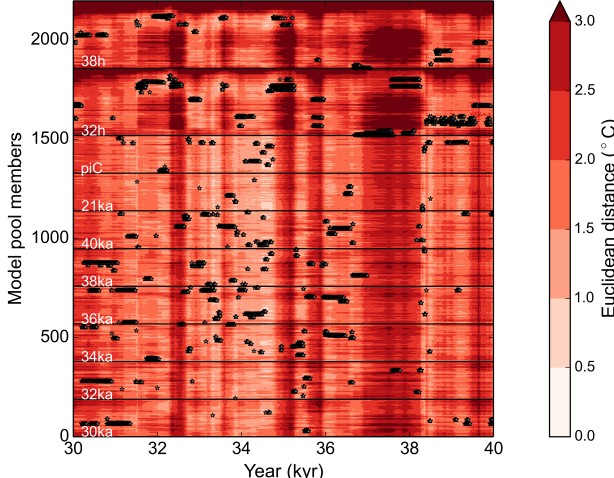

**Figure 4.** Colors show the RMSE between the main model pool and the SST-proxy data. Black dots indicate the ten model years (analogs) that constitute the composite for a given year in the surrogate reconstruction (i.e., the years with smallest RMSE). Model members and respective simulations are indicated on the y-axis and in white writing, while the proxy time steps are indicated on the x-axis.





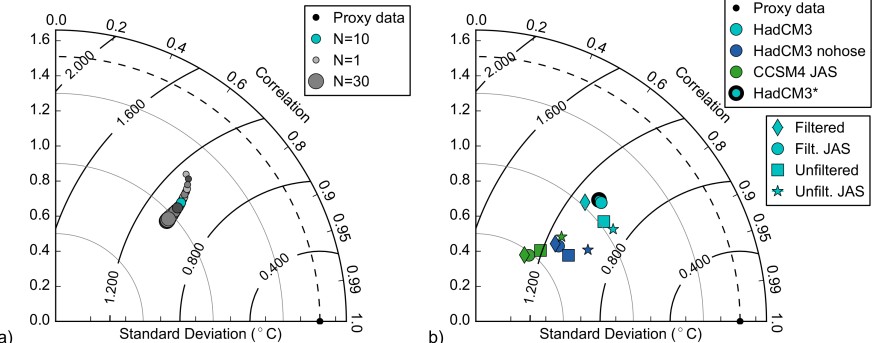

**Figure 5.** Taylor diagram showing the agreement between the proxy data and the surrogate time series produced using a) different number of analogs (1-30) using the main HadCM3 model pool. Increasing size of circles correspond to increasing analog numbers. N=10 is marked in cyan. b) Different model pools (legend). Dashed, black line indicates the standard deviation of the proxy data. Full, black lines indicate the centered RMS difference (labels, in °C). All values are means over the 14 core locations.





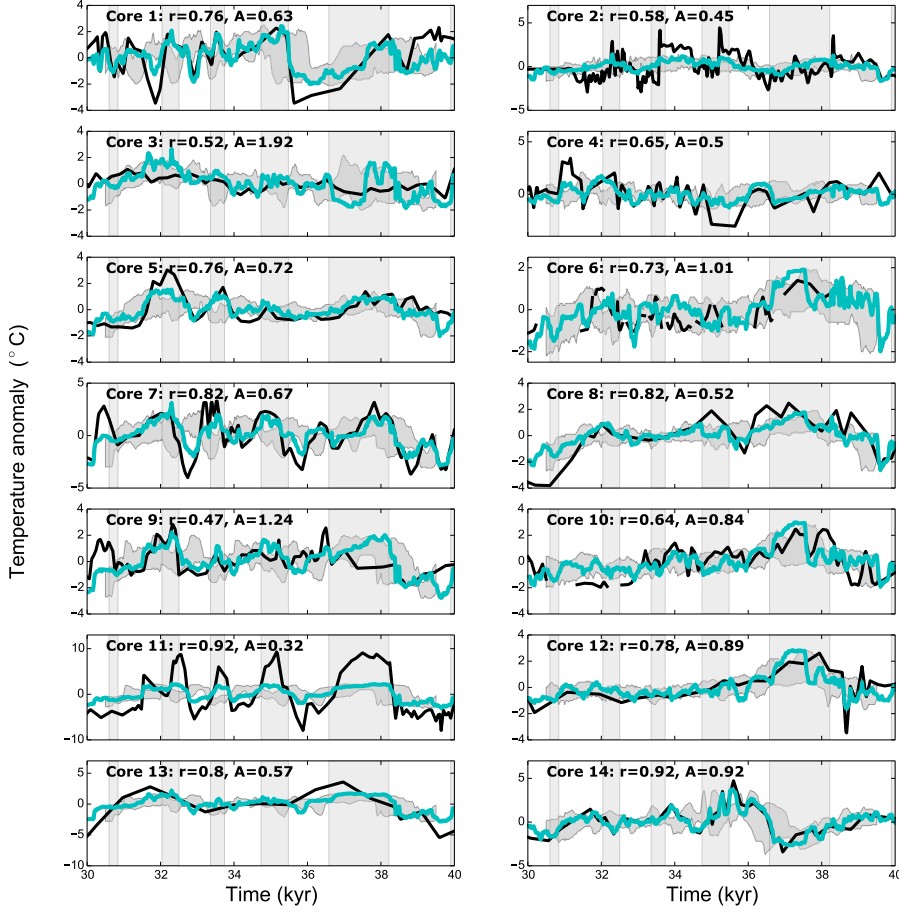

**Figure 6.** Surrogate time series (cyan) and proxy time series (black) for all 14 core locations. All time series are plotted as anomalies from the mean value of the respective time series. The filtered, JAS averaged SSTs from HadCM3 is used. Core numbers correspond to those shown in Fig. 1 and $r$ is the correlation coefficient of the time series at each location. $A$ is the ratio between the standard deviation of the two shown time series, $A=\sigma(c_i)/\sigma(T_i^p)$. The grey shading around the surrogate time series is the 90% confidence interval produced by shifting each proxy record by $\pm$ 500 years. The vertical bars represent interstadial conditions on Greenland as defined by Rasmussen et al. (2014).



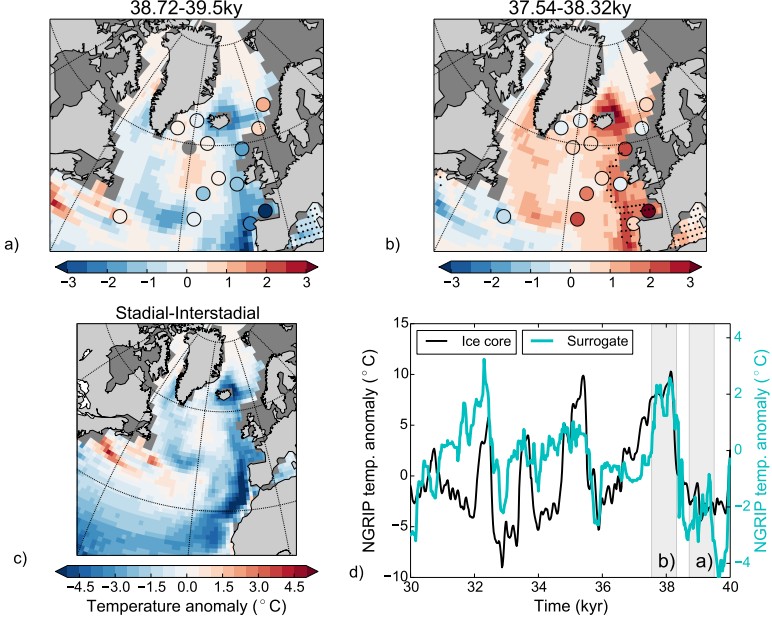

**Figure 7.** Temporal means of the spatial patterns for a) end of GS9, b) beginning of GI8, and c) GS9-GI8. Background color show the temperature anomalies from the surrogate reconstruction, while colors in circles indicate temperature anomalies from the proxy data (missing values for core 5 in the time-period of a)). Black (yellow) stars mark points where the surrogate reconstruction using the original age models is not within the 90% confidence level using age offsets of 500 years (dropping each core, all values within). Panel d) shows reconstructed NGRIP temperatures from the ice core (black line, right y-axis, Kindler et al., 2014) and the values from the surrogate time series (blue, note different y-axis). The surrogate reconstruction is made using the same analog-years which were picked for the SST reconstruction. The grey, vertical shadings show the time periods used in the other panels.





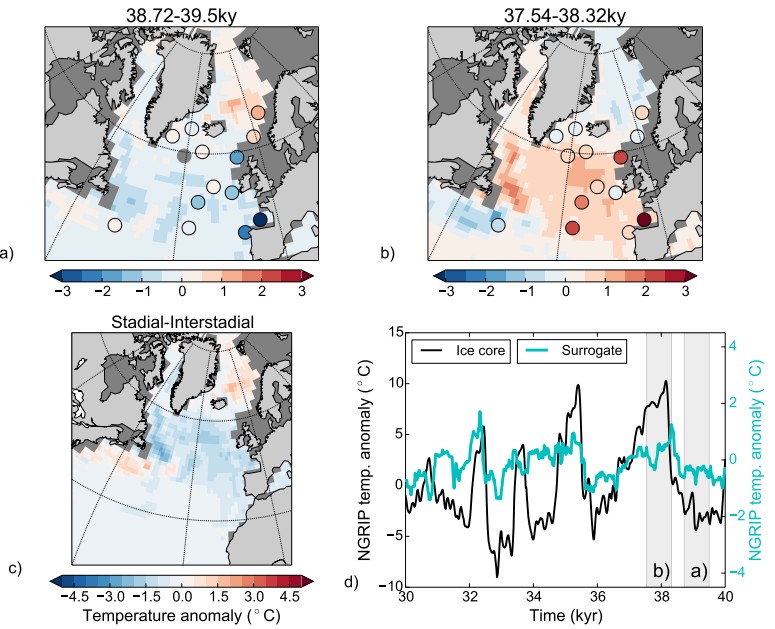

**Figure 8.** Fig. 7 but for HadCM3 nohose.





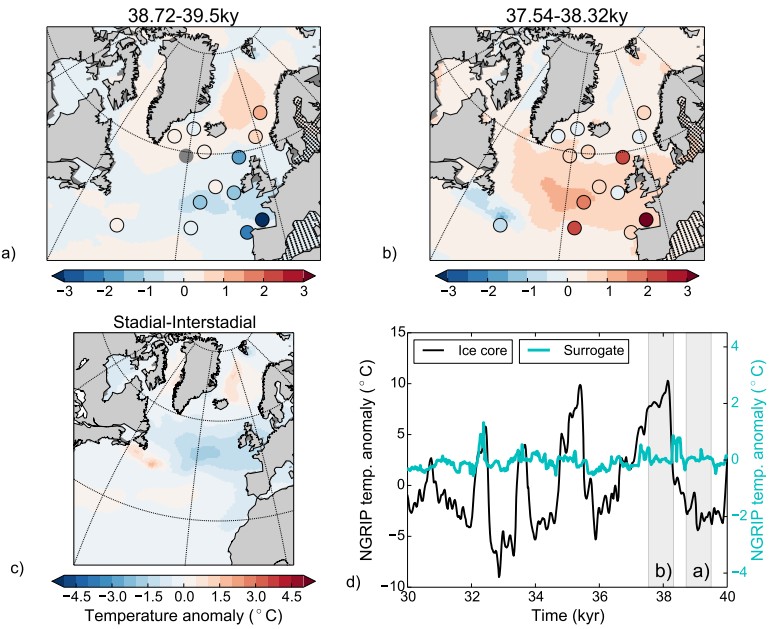

**Figure 9.** Fig. 7 but for CCSM4 data.




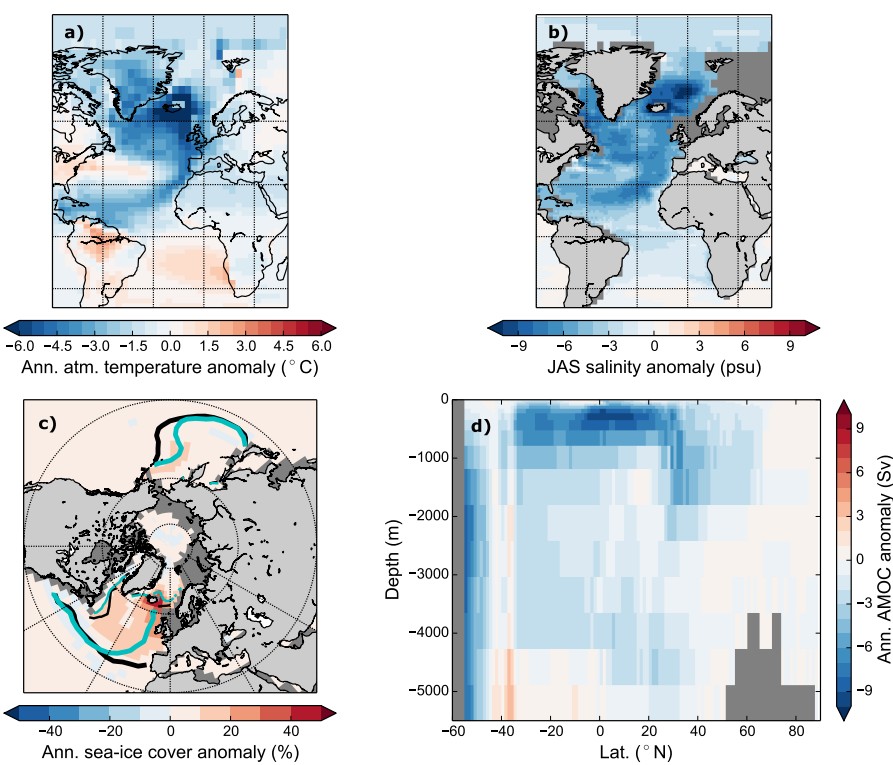

**Figure 10.** Composite of stadial-interstadial conditions (38.72-39.5 kyr vs. 37.54-38.32 kyr) for the surrogate time series constructed using the analogs picked with SSTs from the HadCM3 pool for a-c and HadCM3⋆ for d. Colors show a) annual atmospheric temperature anomalies, b) JAS ocean salinity anomalies, c) annual sea-ice cover concentration anomalies, and d) annual AMOC anomalies. The black and blue contour lines in panel c are the 0.15 sea-ice concentration lines for stadial and interstadial, respectively, thick for March, thin for September.





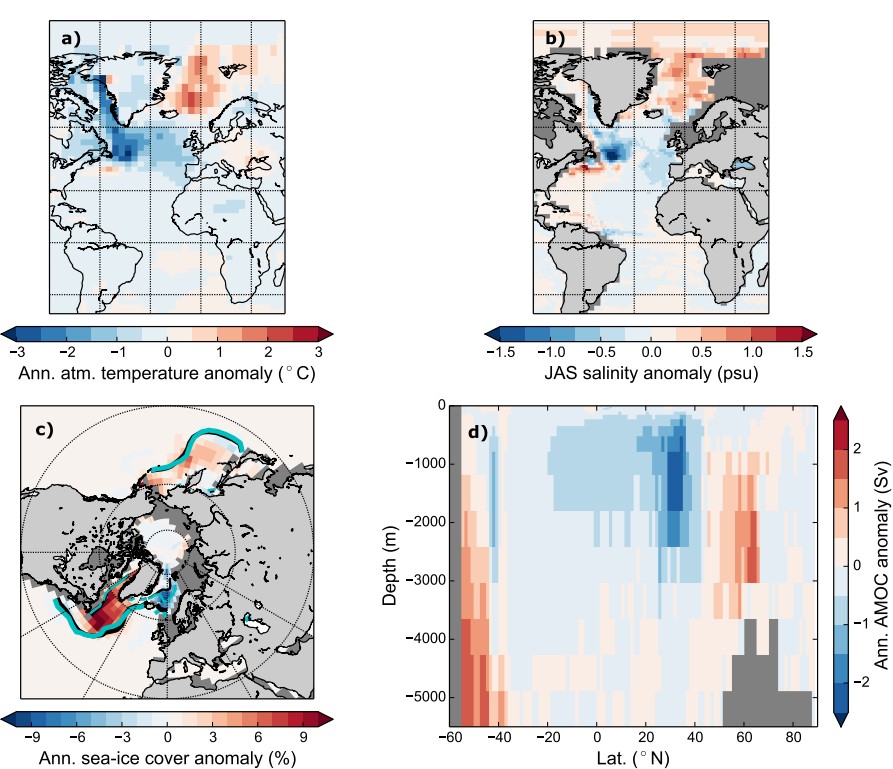

**Figure 11.** Fig. 10 for the HadCM3 nohose model pool





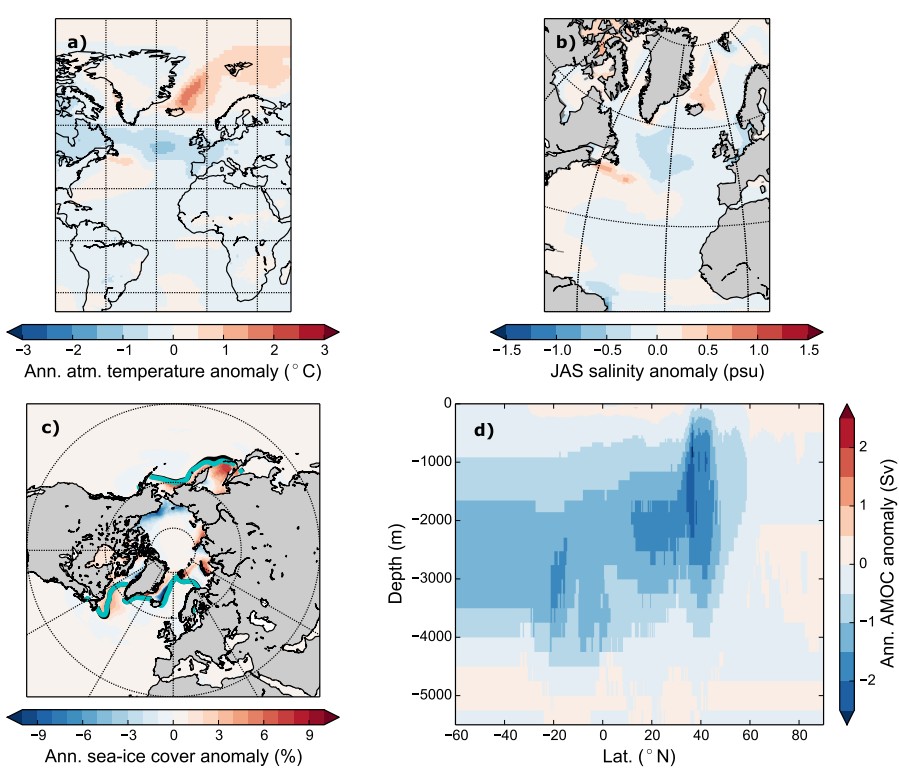

**Figure 12.** Fig. 10 with the CCSM4 model pool. For clarity, only the contours of the March sea-ice extent are shown.



**Table 1.** Core sites and age model information

| Core nr. | Name | Method | Basis for age model 30–40 kyr | Reference |
|---|---|---|---|---|
| 1 | MD95-2010 | ML | Two age models presented and discussed. 1) Calibrated $^{14}$C dates (9), and NAAZ II and Laschamp magnetic excursion. 2) Magnetic Susceptibility (MS) vs. GISP2 $\delta^{18}$O. The deviation between the $^{14}$C based and the MS vs. GISP2 $\delta^{18}$O age models are always less than 1 kyr and mostly close to zero | *Unpub. data T. Dokken* Age model: Dokken and Jansen (1999) |
| 2 | MD99-2284 | ML | Age model are based on Anhysteretic Remanent Magnetism vs NGRIP $\delta^{18}$O and 2 ash horizons (FMAZ III and II). The age model is supported by calibrated $^{14}$C dates (5) that are not used as tie points in the established age model, a well as the Laschamp and Mono Lake magnetic excursions | Dokken et al. (2013) |
| 3 | JM96-1225 | ML | Calibrated $^{14}$C dates (2) and ash zone II (at 52 ka) | Hagen and Hald (2002) |
| 4 | SU90-24 | ML | Uncalibrated $^{14}$C dates (7) | Elliot et al. (1998) |
| 5 | ODP 983 | %NP | Warming at the site are assumed to be synchronous with warming of the wider North Atlantic and these warm events are aligned to a synthetic Greenland record at EDC3 age model. | Barker et al. (2015) |
| 6 | SO82-5-2 | %NP | Age model based on the combination of information given by $\delta^{18}$O, $^{14}$C dates (28), and tuning of SSTs to GISP2 $\delta^{18}$O | van Kreveld et al. (2000) |
| 7 | MD04-2829CQ | ML | Age model based on calibrated $^{14}$C dates (5), followed by a fine tuning of the %NP, used a SST indicator, to GISP2 $\delta^{18}$O. The Lachamp magnetic excursion is identified | Hall et al. (2011) |
| 8 | GIK23415-9 | ML | Age model based on correlation of IRD and $\delta^{18}$O to GISP2 and supported by the calibrated $^{14}$C dates (6) | Weinelt et al. (2003), Weinelt (1993) |
| 9 | MD01-2461 | %NP | Age model based on calibrated $^{14}$C dates (4), followed by a fine tuning of the %NP, used a SST indicator, to GISP2 $\delta^{18}$O. Ash horizons. | Peck et al. (2007) |
| 10 | DSDP 609 | %NP | Age model based on tuning the %NP, used a SST indicator, to GISP2 $\delta^{18}$O | Bond et al. (2013) |
| 11 | MD04-2845 | ML | Identified regional climatic and biostratigraphic events are correlated to the same events identified in MD04-2042. MD04-2042 is tuned to GISP2 $\delta^{18}$O following the assumption that Greenland temperature changes associated with DO-events were synchronous over the North Atlantic region (Bard et al., 2004). Supported by calibrated $^{14}$C dates (3). The not used $^{14}$C dates deviate from the established age model by 90-1250 years. | Sánchez Goñi et al. (2008) |
| 12 | GIK15612-2 | ML | IRD maxima related to Heinrich layers are tuned to GISP2 $\delta^{18}$O. The chronology is supported by calibrated $^{14}$C dates not used to establish the age model | Kiefer (1998) |
| 13 | SU92-03 | ML | Age model based on tuning $\delta^{18}$O$_p$ and %NP to GISP2 $\delta^{18}$O and Heinrich layers. The age model is also compared to MD95-2040 (de Abreu et al., 2003) and MD95-2042 (Shackleton et al., 2000), where also $\delta^{18}$O$_p$ and SSTs were tuned to GISP2 $\delta^{18}$O | Salgueiro et al. (2010) |
| 14 | CH69-K09 | ML | Calibrated $^{14}$C dates (6) | Labeyrie et al. (1999), Waelbroeck et al. (2001) |

Numbers in parenthesis mark number of $^{14}$C dates. FMAZ: Faeroe Marine Ash Zone. NAAZ: North Atlantic Ash Zone. IRD: ice-rafted debris



**Table 2.** GCM simulations

| Name | Simulated years | Period | Horz. resolution ocean | Hosing | Reference |
|------|------|------|------|------|------|
| HadCM3 - tcmqw3 | 200 (⋆:100) | 30 ka | 1.25° × 1.25° | NO | Singarayer and |
| HadCM3 - tcmob3 | 200 (⋆:100) | 32 ka | 1.25° × 1.25° | NO | Valdes (2010) |
| HadCM3 - tcmqv3 | 200 (⋆:100) | 34 ka | 1.25° × 1.25° | NO | |
| HadCM3 - tcmoa3 | 200 (⋆:100) | 36 ka | 1.25° × 1.25° | NO | |
| HadCM3 - tcmqu3 | 200 (⋆:100) | 38 ka | 1.25° × 1.25° | NO | |
| HadCM3 - tcmpu3 | 200 (⋆:100) | 40 ka | 1.25° × 1.25° | NO | |
| HadCM3 - tcmoe3 | 200 (⋆:100) | LGM | 1.25° × 1.25° | NO | |
| HadCM3 - tcmfa3 | 200 (⋆:100) | PI | 1.25° × 1.25° | NO | |
| HadCM3 - tcoxj+ | 99 hosed year + 250 non-hosed years | 32 ka / 32 ka | 1.25° × 1.25° | YES | This study + Singarayer and Valdes (2010) |
| HadCM3 - tcoxk+ | 99 hosed year + 250 non-hosed years | 38 ka / 32 ka | 1.25° × 1.25° | YES | |
| CCMS4 PI-1 | 1000 | PI | Bipolar 1° | NO | Gent et al. (2011), Danabasoglu et al. (2012) |
| CCMS4 PI-2 | 1000 | PI | Bipolar 1° | NO | Kleppin et al. (2015), Born and Stocker (2014) |
| CCMS4 LGM | 101 | LGM | Bipolar 1° | NO | Brady et al. (2013) |

| Model pool | Simulations | | | | |
|------|------|------|------|------|------|
| **HadCM3 (main)** | tcmqw3, tcmpu3, | tcmob3, tcmoe3, | tcmqv3, tcmfa3, | tcmoa3, tcoxj+, | tcmqu3, tcoxk+ |
| **HadCM3 nohose** | tcmqw3, tcmpu3, | tcmob3, tcmoe3, | tcmqv3, tcmfa3 | tcmoa3, | tcmqu3, |
| **CCSM4** | PI-1, | PI-2, | LGM | | |
| **HadCM3⋆** | tcmqw3⋆, tcmpu3⋆, | tcmob3⋆, tcmoe3⋆, | tcmqv3⋆, tcmfa3⋆, | tcmoa3⋆, tcoxj+, | tcmqu3⋆, tcoxk+ |





**Table 3.** The agreement between the PSR and the ice-core record

| Ice core | NGRIP | | | GISP2 | | |
| --- | --- | --- | --- | --- | --- | --- |
| Model pool | $r$ | $r_{36-40\mathrm{kyr}}$ | A | r | $r_{36-40\mathrm{kyr}}$ | r |
| **HadCM3** | 0.49 | 0.81 | 0.35 | 0.63 | 0.84 | 0.32 |
| HadCM3 nohose | 0.47 | 0.66 | 0.13 | 0.63 | 0.76 | 0.13 |
| CCSM4 | 0.25 | 0.13 | 0.06 | 0.31 | 0.21 | 0.06 |