# Peer review of "A spatio-temporal reconstruction of sea-surface temperatures in the North Atlantic during Dansgaard-Oeschger events 5-8"

_Climate of the Past, 2017_

## Referee Comment (RC1) · J.J. Gómez-Navarro (Referee) · 18 Sep 2017

I have reviewed the article entitled "A spatio-temporal reconstruction of sea-surface temperatures in the North Atlantic during Dansgaard-Oeschger events 5–8" by M. F. Jensen et al. The article performs a Proxy Surrogate Reconstruction based on a number of proxies from marine sediments and SST fields borrowed from coupled atmosphere-ocean climate model simulations. The authors evaluate the performance of the methodology and assess their main uncertainties. The final product enables the authors exploring some physical mechanisms that might be important in previous DO events.

General Comment:

Although the methodology is not entirely new, it is to my knowledge the first time that this technique has been applied to reconstruct climate states so far from current conditions. Uncertainties are large and populate most aspects of the methodology that limit the robustness of the findings. Still, the method seems reasonably robust and the conclusions coherent. With a healthy level of caution, the methodology has the potential to shed light into complex and important physical mechanisms that remain poorly understood. The structure is sensible, the selection of figures adequate, and the text is easy to read. For these reasons, I think the manuscript merits being published, perhaps with few minor corrections that in my opinion could help to clarify some details of the methodology I found a bit confusing when reading it the fist time.

Specific comments:

- The abstracts reads a bit optimistic regarding the uncertainties. Contrary to the Conclusions section, it ends highlighting the robustness, rather than the limitations and this being an 'encouraging fist attempt'. In my opinion, the conservative tone of the conclusions (e.g. Page 14, Lines 25-27) could be reproduced as well in the abstract.

- The paragraph in the Introduction starting in Page 3, Line 26 enumerates conclusions. Therefore, it does not introduce the problem, and I think this text should be moved to/merged with the final section

- The three first lines in Section 2.1 can be merged with the following paragraph to avoid such short one. Also, it is a bit misleading, as it is not clear whether it was Lorenz or Graham, the first study introducing the PSR. What "approach" exactly introduced Lorenz? (By the way, I know the answer: Lorenz introduced the analog method, while Graham introduced PSR)

- In Page 4, expressions in parenthesis such as "see below" "see Sect..." are a bit obvious and might be removed. Also, it reads "A collection of years from one or several

model simulations are treated as pool..." I think this is not entirely correct. As far as I understood, a collection of years is used, together with a low-pass filter, to BUILD a climate state. The method is based on the search across such "climate states", rather than on individual years. Am I right? If so, please clarify. If not, please clarify.

- Page 4, Lines 13-15: Be careful with this argument. Although in principle the method gives you a date, and you can draw any variable concurrent with that date from the GCMs pool, this does not mean that any variable can be reconstructed. Only those variables that are physically and strongly connected with SST, which is the only variable constrained by the proxy information, can be reconstructed to a certain degree. This pertains for example the test done with NGRIP temperatures in Section 4.3, which by the way could perhaps be evaluated using only synthetic data in Section 3.1.

- The name "Data pool" in section 2.2 is a bit misleading. Firstly, because models are also data, although they are not described here. Secondly, because I'm not sure if "pool" is an adequate naming for a set of marine records. I would call it "Proxy records" or something similar. Note however that this does not apply to section 2.3, whose name "Model Pool" is in its context accurate and descriptive.

- Page 6, Lines 19-20: How exactly is such a test carried out? Where are the results shown? I guess this pertains the gray shadow in Fig. 6 Is it so? I could not understand the details of how such a uncertainty interval was obtained.

- Page 6. Regarding the simulation, only 200 years out of the 500 available are used. Why so? (This also applies to Line 8 in Page 7) Given that the pool size is critical for analog-like reconstructions, wouldn't it be better to keep as many years as possible? Even if such years are less reliable individually (for spinup considerations, I imagine), their inclusion in the pool enlarges its variability with more heterogeneous "climate states". Therefore, it can only enrich the pool with heterodox, unlikely states which might be, perhaps even by chance, more adequate to reproduce the exceptional situations under DO events.

[Figure]

- Page 7, Lines 5. The term "8 unforced simulations" is misleading. Here, it refers to the fresh water only, but readers might think that it refers to absence of forcings at all, including orbital, GHG, solar, etc. This is, "unforced" seems to mean pure control simulations, which as far as I understood is not what these simulations consists of. Note that this remark applies to many instances through the manuscript. I would advise reviewing every instance of the word "forcing" in the text and reword it accordingly to clarify that the term reefers to "normal salt concentration", but the rest of the variables being normally forced.

- As a general note regarding Section 2.3, it would be nice to sum up how many years there are in total available within the pool. Anyway, it seems clear that there are fewer years than those that are being reconstructed. This is a rather undesirable situation (normally there are many more analogs available than necessary, and some authors still complain about the small size of the pool given the large dimensionality of the problem). Although I understand that this is unavoidable, perhaps it is worth to point this out, adding few words of caution that at least demonstrate that the authors are aware of this fact.

- Page 7, Lines 25- Why only 30? Are they continuous periods? Or are they 30 random, fully disconnected samples? In any case, why such a choice?

- I'm not sure if the design of the experiment in Section 3.1 is the best one. It uses the same model to produce the target and draw analogs. It does not even contaminate the synthetic proxies to mimic a more realistic scenario. Therefore, the results suppose a very optimistic upper bound, barely representative of the actual performance of the technique. With the available data, it is rather easy to design more illustrative experiments which lead to tighter bounds of the uncertainty: adding noise or using a different model (e.g. CCSM4) as target are some examples.

- Page 8, Line 11. The Fig. 4 is barely explained. After having read the paper a couple of times, I still do not fully understand it. I think more details should be provided.

- Page 8, Line 20. How is this correlation/standard deviation calculated? Over the 14 locations? Are the values shown in the Figure the results of such cuantities averaged over the full period? I think this part lacks details that facilitate the read of the conclusions.

- Page 8, Lines 27-30: I'm not sure if the inclusion of the "unfiltered data" is necessary. Such data is not described before (only averaged climate states are expected at this point), therefore it is a bit misleading. The test is in any case incorrect, as it makes little sense to compare proxy data (representative of low-pass frequency) with yearly averages, and the conclusion are rather trivial (obviously the year-to-year variability is larger than the low-pass filtered). Therefore, I think it would be better to remove such test from the text and Fig. 5 for the sake of clarity.

- Page 10, Line 13-15: Again, how is this interval exactly calculated. In Fig. 2. shouldn't the black line be included within the gray shadow?

- Fig 1: The figure lacks a colour scale.

- Fig. 2: Why are there gaps in some cores (6, 10)?

- Fig 4: The label reads "Euclidean distance", rather than RMSE. This figure is hard to understand, and further details could be added to facilitate its read. For instance, are there 10 black dots per column? What is the number of rows? Further, this figure exhibits a lot of structure. The black dots are far from homogeneously distributed, which in my opinion deserves some more attention that the one demonstrated in the text.

---

## Short Comment (SC1) · 26 Sep 2017

The PAGES Data Stewardship Integrative Activity seeks to advance best practices for sharing the data generated and assembled as part of all PAGES-related activities. The CP Special Issue, "PAGES Young Scientists Meeting 2017" is part of this PAGES activity. The co-editors of the Special Issue are reviewing the data availability within each of the CP-Discussion papers in relation to the CP data policy (https://www.climate-of-the-past.net/about/data_policy.html) and current best practices. The editor team is making recommendations for each paper, with the goal of achieving a high and consistent level of data stewardship across the Special Issue. We recognize that an additional effort

will likely be required to meet the high level of data stewardship envisaged, and we appreciate the dedication and contribution of the authors. This includes the use of Data Citations (see example below). Authors are also strongly encouraged to deposit significant code into a suitable repository and to cite it using a Data Citation.

We ask authors to respond to our comments as part of the regular open interactive discussion. If you have any questions about PAGES Data Stewardship principles, please contact any of us directly. Best wishes for the success of your paper.

YSM Special Issue editor team

D.S. Kaufman, M.F. Loutre, M.N. Evans, S.C. Fritz, C. Tabor, H. Plumpton, R. Barnett, Y. Zhang, E. Razanatsoa, and E. Dearing Crampton Flood

————

For this paper: (1) Research input data – selected 14 marine sediment cores (a) Include Data Citations or URL or doi links (not just bibliographic references) for the foraminifera assemblages behind each of the 14 proxy records listed in Table 1. (b) Submit the new SST reconstructions, including estimates of uncertainty, to a public repository. If the assemblage data have not been archived before, the authors should work with the data generators to obtain a Data Citation with a persistent identifier (URL or doi) for the dataset. (c) Add a Data Citation or URL link for access to the SST calibration dataset. It is not discoverable through the currently cited Kucera et al., 2005 paper.

(2) Research input data – climate model simulations Deposit the essential climate model simulations that have not already been archived into a trusted data repository and include the link or citation in Table 2. For CCSM4 simulations, please add a sentence to the Data Availability section with instructions on how to access the data.

(3) Research output data – SST field reconstructions Apart from the new site-level SST reconstructions (above Comment 1), the primary contribution of this study is the

spatially distributed SST reconstruction over the North Atlantic as illustrated at time slices using different model simulations in Figs 7-9. For intelligent reuse of such primary results, please submit to a public repository the gridded SST values over the reconstruction domain at some reasonable time step (including DI-8 and DS-9).

(4) Research output data – other climate variables In addition to the SST field reconstruction (above), this study includes an analysis of other climate variables that are linked to SST, namely sea-ice cover and AMOC. We encourage the authors to deposit these datasets, which were used to draw conclusions about these climate variables along with the SST field reconstruction.

————-

What is a "Data Citation"? Data Citations track the provenance of a dataset giving credit to the data generator; this is in addition to any references to publications where the data are described. Data Citations are used in the text (or tables) alongside and in the same way as publication citations. In the Reference list, they include: Creators, Title, Repository, Identifier, Submission Year. More information about Data Citations is here: <https://www.datacite.org/mission.html> Here is an example of text and corresponding citations (using CP punctuation style):

"The PAGES2k Consortium (2017a) assembled a large global dataset of temperature-sensitive proxy records (PAGES2k Consortium, 2017b). Among the records is the paleo-temperature reconstruction from Laguna Chepical (de Jong et al., 2016), which was described by de Jong et al. (2013)."

References

de Jong, R., von Gunten, I., Maldonado, A., and Grosjean, M.: Late Holocene summer temperatures in the central Andes reconstructed from the sediments of high-elevation Laguna Chepical, Chile (32° S), Climate of the Past, 9, 1921-1932, 2013.

de Jong, R., von Gunten, I., Maldonado, A., and Grosjean, M.: Laguna Chepical summer temperature reconstruction, World Data Center for Paleoclimatology, https://www.ncdc.noaa.gov/paleo/study/20366, 2016.

PAGES 2k Consortium: A global multiproxy database for temperature reconstructions of the Common Era, Scientific Data, 4,170088, 2017a.

PAGES 2k Consortium: A global multiproxy database for temperature reconstructions of the Common Era, version 2.0.0, figshare, https://figshare.com/s/d327a0367bb908a4c4f2, 2017b.

---

## Short Comment (SC2) · 6 Oct 2017

Thank you for your comment. We are working on the mentioned points, and provide some short, preliminary comments below.

1. For the research input data, the 14 marine sediment cores, we are working on providing Data Citations/links and uploading data. a) We will provide links for the faunal data in Tab. 1 b) We will upload the new SST reconstructions to a public repository c) We will provide a Data Citation to the SST calibration dataset

2. Regarding the climate model simulations. The CCSM4 simulations are already

publicly available, and we will add a sentence to the Data Availability section on how to access data. We are looking into the HadCM3 model simulations. However, each 500 year simulation has about 100GB of data associated with it, resulting in terabytes of output for all the simulations. This would easily fill up a database and would be very expensive. The simulations can be browsed through http://www.bridge.bris.ac.uk/resources/simulations and are also available upon request.

3&4: We are looking into uploading the research output data

———————————————————

---

## Author Comment (AC1) · 6 Oct 2017

We thank the referee for his review and suggestions to improvements. Generally, we agree with all his suggestions and think they will help to clarify the manuscript. We provide a comment (marked with C) to all specific comments (reproduced, marked with R) below. The manuscript will be updated accordingly when the full review is available.

——————————————————————————————————————————-

R: The abstracts reads a bit optimistic regarding the uncertainties. Contrary to the Conclusions section, it ends highlighting the robustness, rather than the limitations and

this being an 'encouraging fist attempt'. In my opinion, the conservative tone of the conclusions (e.g. Page 14, Lines 25-27) could be reproduced as well in the abstract.

C: We agree with the referee and will change the tone of the abstract in the revised manuscript

———————————————————————————————————————————-

R: The paragraph in the Introduction starting in Page 3, Line 26 enumerates conclusions. Therefore, it does not introduce the problem, and I think this text should be moved to/merged with the final section

C: We will move and merge the paragraph with the final section

———————————————————————————————————————————-

R: The three first lines in Section 2.1 can be merged with the following paragraph to avoid such short one. Also, it is a bit misleading, as it is not clear whether it was Lorenz or Graham, the first study introducing the PSR. What "approach" exactly introduced Lorenz? (By the way, I know the answer: Lorenz introduced the analog method, while Graham introduced PSR)

C: We will merge the two paragraphs and make clear that Lorenz introduced the analog method

———————————————————————————————————————————-

R: In Page 4, expressions in parenthesis such as "see below" "see Sect..." are a bit obvious and might be removed. Also, it reads "A collection of years from one or several model simulations are treated as pool..." I think this is not entirely correct. As far as I understood, a collection of years is used, together with a low-pass filter, to BUILD a climate state. The method is based on the search across such "climate states", rather than on individual years. Am I right? If so, please clarify. If not, please clarify.

C: The three "see-expressions" in parenthesis in Sec. 2.1 will be removed. We will

make clear that the search is done across the "built climate states" (you are right). "A low-passed filtered collection of years from one or several model simulations are treated as a pool of possible "climate states"

––––––––––––––––––––––––––––––––––––––––––––––––––––––––––––––––––-

R: Page 4, Lines 13-15: Be careful with this argument. Although in principle the method gives you a date, and you can draw any variable concurrent with that date from the GCMs pool, this does not mean that any variable can be reconstructed. Only those variables that are physically and strongly connected with SST, which is the only variable constrained by the proxy information, can be reconstructed to a certain degree. This pertains for example the test done with NGRIP temperatures in Section 4.3, which by the way could perhaps be evaluated using only synthetic data in Section 3.1.

C: We will change the wording of the argument, thanks for pointing this out. We will also evaluate the agreement between NGRIP temperatures in the synthetic test. The correlation between temperatures at NGRIP from the PSR with synthetic data and the synthetic data (CCSM4 output as requested below) is 0.75, showing a connection with SSTs.

––––––––––––––––––––––––––––––––––––––––––––––––––––––––––––––––––-

R: The name "Data pool" in section 2.2 is a bit misleading. Firstly, because models are also data, although they are not described here. Secondly, because I'm not sure if "pool" is an adequate naming for a set of marine records. I would call it "Proxy records" or something similar. Note however that this does not apply to section 2.3, whose name "Model Pool" is in its context accurate and descriptive.

C: We will change the section name to "Proxy records"

––––––––––––––––––––––––––––––––––––––––––––––––––––––––––––––––––-

R: Page 6, Lines 19-20: How exactly is such a test carried out? Where are the results shown? I guess this pertains the gray shadow in Fig. 6 Is it so? I could not understand

the details of how such a uncertainty interval was obtained.

C: We acknowledge that this test was not made clear in the manuscript and will change that in the revised manuscript.

We change the age model of each proxy record individually by +-500 years, thus giving $2^{14}$ (14 proxy records) different perturbations for the entire set of proxy records. We find a PSR for each option, giving $2^{14}$ different PSRs. The grey shading in Fig. 6 is the 90% confidence interval from all those PSRs. The results are also shown in Fig 7, where the black dots show where the PSR using the original age models do not fall within the same 90% confidence interval. I.e., grid cells without a black dot has a temperature anomaly that falls within the 90% confidence interval produced with the $2^{14}$ PSRs. The dots are also included in Fig 9 for the CCSM4-model pool, but are missing in Fig 8 and we will test the HadCM3 nohose model pool accordingly. We note that because of the very large number of perturbations, it was technically infeasible to keep the full data for all $2^{14}$ PSRs. We thus saved the results in bins of size 0.1°C.

—————————————————————————————————————————-

R: Page 6. Regarding the simulation, only 200 years out of the 500 available are used. Why so? (This also applies to Line 8 in Page 7) Given that the pool size is critical for analog-like reconstructions, wouldn't it be better to keep as many years as possible? Even if such years are less reliable individually (for spinup considerations, I imagine), their inclusion in the pool enlarges its variability with more heterogeneous "climate states". Therefore, it can only enrich the pool with heterodox, unlikely states which might be, perhaps even by chance, more adequate to reproduce the exceptional situations under DO events.

C: We agree that the pool size should be maximized. Unfortunately, not all years are easily available. Line 8, Page 7: The remaining years were not available through the CCSM4 database, and so therefore, availability is the reason for not including these years. Page 6: We are currently checking whether the remaining 300 years of the

HadCM3 data pool are available. If so, we aim to perform a test with all years included to see whether this changes the result and if the spin-up years are chosen. The original reason for not including those years are spinup considerations, as you mention. The simulations are a model drift from an initial (pre-ind) climate state towards MIS3/LGM model climate. Including the spin-up years might be problematic as they may be influenced by unrealistic initial conditions (e.g., ocean at rest, homogeneous temperatures and salinities), in this case a control PI-simulation. While it may be interesting to do and (hopefully) show that we don't pick these states, the information they would yield would simply be too unreliable, and therefore we can a priori discard these as unfit for the method.

————————————————————————————————————————————-

R: Page 7, Lines 5. The term "8 unforced simulations" is misleading. Here, it refers to the fresh water only, but readers might think that it refers to absence of forcings at all, including orbital, GHG, solar, etc. This is, "unforced" seems to mean pure control simulations, which as far as I understood is not what these simulations consists of. Note that this remark applies to many instances through the manuscript. I would advise reviewing every instance of the word "forcing" in the text and reword it accordingly to clarify that the term reefers to "normal salt concentration", but the rest of the variables being normally forced.

C: We will point out that we are talking about freshwater forcing, and not the boundary conditions. However, note that each of the "8 unforced simulations" consist of constant orbital, GHG and solar forcings. Differences only exists from simulation to simulation, not within, and the experiments are not necessarily more forced than a pre-industrial simulation.

————————————————————————————————————————————-

R: As a general note regarding Section 2.3, it would be nice to sum up how many years there are in total available within the pool. Anyway, it seems clear that there

are fewer years than those that are being reconstructed. This is a rather undesirable situation (normally there are many more analogs available than necessary, and some authors still complain about the small size of the pool given the large dimensionality of the problem). Although I understand that this is unavoidable, perhaps it is worth to point this out, adding few words of caution that at least demonstrate that the authors are aware of this fact.

C: We will point out the undesirable ratio between years in model pool and years reconstructed and sum up the number of years available within each pool. We try to reconstruct 501 points across 10 000 years using model pools with a size of 2198, 1520 or 2071 years which is clearly not enough to represent the whole interval of 30-40 ka. Although the 501 points are not individual years, the ratio between the points reconstructed and the model pool is more desirable. We did perform a principal component analysis of the proxy record and the HadCM3 model pool, showing that in both cases 9 principal components are needed to explain more than 90% of the temperature variability. This suggest that both the proxy records and the HadCM3 model pool have approximately 9 degrees of freedom.

————————————————————————————————————————————————

R: Page 7, Lines 25- Why only 30? Are they continuous periods? Or are they 30 random, fully disconnected samples? In any case, why such a choice?

C: 30 random, fully disconnected samples. This number was chosen to be small enough to perform a rapid test, but larger than the low-pass filter. We performed a test with 501 random samples in addition (number of proxy time steps to reconstruct). The results are comparable, but as we present the results as averages over the samples, the 95% and 5% offsets get "averaged out" and smaller when 501 samples are used. Therefore, 30 samples seem like a more honest test.

————————————————————————————————————————————————

R: I'm not sure if the design of the experiment in Section 3.1 is the best one. It uses the same model to produce the target and draw analogs. It does not even contaminate the synthetic proxies to mimic a more realistic scenario. Therefore, the results suppose a very optimistic upper bound, barely representative of the actual performance of the technique. With the available data, it is rather easy to design more illustrative experiments which lead to tighter bounds of the uncertainty: adding noise or using a different model (e.g. CCSM4) as target are some examples.

C: We will change the target to CCSM4 data and keep the HadCM3 as the model pool. This was already done, but the results only briefly mentioned. The results are attached in Figure1 which shows the same as the Figure 3 in the manuscript, but with CCSM4 as the target (re-gridded to the HadCM3 grid). The resulting offsets between the PSR and the original CCSM4 values are similar, but the structure is different. We note that the correlation decreases outside the core-locations. We will update the manuscript accordingly.

———————————————————————————————————————————-

R: Page 8, Line 11. The Fig. 4 is barely explained. After having read the paper a couple of times, I still do not fully understand it. I think more details should be provided.

C: We will provide more details. Comments are provided in the last point of this review.

———————————————————————————————————————————-

R: Page 8, Line 20. How is this correlation/standard deviation calculated? Over the 14 locations? Are the values shown in the Figure the results of such cuantities averaged over the full period? I think this part lacks details that facilitate the read of the conclusions.

C: We will provide more details. The quantities in Fig 5 is averaged over the full period and then over the 14 core locations

———————————————————————————————————————————-

R: Page 8, Lines 27-30: I'm not sure if the inclusion of the "unfiltered data" is necessary. Such data is not described before (only averaged climate states are expected at this point), therefore it is a bit misleading. The test is in any case incorrect, as it makes little sense to compare proxy data (representative of low-pass frequency) with yearly averages, and the conclusion are rather trivial (obviously the year-to-year variability is larger than the low-pass filtered). Therefore, I think it would be better to remove such test from the text and Fig. 5 for the sake of clarity.

C: We will remove those tests from Fig. 5 and only show diamonds+ circles (annual + JAS)

––––––––––––––––––––––––––––––––––––––––––––––––––––––––––––––––––––-

R: Page 10, Line 13-15: Again, how is this interval exactly calculated. In Fig. 2. shouldn't the black line be included within the gray shadow?

C: The intervals are the results from Fig. 6, r and A calculated for the full period for each core location. We will make it more clear. For the last part of the comment, do you mean Fig. 6? Black line should not be within the gray shadow since it is the proxy record, neither should the cyan line, since the gray shadow is the envelope of shifting each proxy +/- 500 years, while the cyan line is the result when nothing is shifted. If the shadow would be done by shifting each record by +/- 500 year by year the cyan would be within the shadow, but this would be too many permutations for us to perform. The shadow gives an envelope that illustrates uncertainty arising from the age models.

––––––––––––––––––––––––––––––––––––––––––––––––––––––––––––––––––––-

R: Fig 1: The figure lacks a colour scale.

C: We will add

––––––––––––––––––––––––––––––––––––––––––––––––––––––––––––––––––––-

R: Fig. 2: Why are there gaps in some cores (6, 10)?

C: Because the cores have %np values outside the range where the linear relationship between temperature and %np holds because the population of this particular species saturates near 100% or decreases to close to 0% (Page 6, line 8). Extrapolation of the temperature reconstruction would lead to very uncertain results that could greatly harm the analog method. On the other hand, unlike other methods the PSR is very robust to missing data of which we take advantage here.

—————————————————————————————————————————————-

R: Fig 4: The label reads "Euclidean distance", rather than RMSE. This figure is hard to understand, and further details could be added to facilitate its read. For instance, are there 10 black dots per column? What is the number of rows? Further, this figure exhibits a lot of structure. The black dots are far from homogeneously distributed, which in my opinion deserves some more attention that the one demonstrated in the text.

C: We will change Euclidean distance to RMSE, and provide more information about the figure itself. There are 10 black dots per column, and 501 per row (every 20 years). There are 2198 rows in total. Black lines separate the different simulations.

The focus in the text is on GS9/GI8 where black dots are more homogenously distributed. However, we do agree that this is not true for the remaining time-interval and will discuss this briefly in the text

—————————————————————————————————————————————-
————————————————————————————

[Figure]

**Fig. 1.**

---

## Referee Comment (RC2) · Anonymous Referee #2 · 23 Jan 2018

The paper by Jensen and co-authors is a very interesting contribution, very well written, which I find particularly useful for the paleoclimate community. The proxy surrogate reconstruction method allows for a better understanding of the climate system beyond the spatial limitations (and variables to be reconstructed) by proxy data. I recommend the manuscript for publication, after a few corrections and some clarifications. Page 5. Data Pool. I assume the authors reconstruct SST at 10 m following methodology by Kucera et al. 2005 within the MARGO framework. However, Telford et al. 2014 demonstrated that this depth rarely is the most significant for fossil planktonic foraminifera assemblages in the North Atlantic, and more sensitive to subsurface conditions. This should be acknowledged by the authors. I may understand that for their purposes, this

is not a very important issues, but it should be at least explained how this may affect the proxy-model comparison. Page 6. L. 22. How the larger age uncertainty for cores 3,4 and 14 may affect the comparison between the surrogate and proxy time series? Could you add some sentence about this? Page 10. L. 13. r instead of r2. Figure 7. Black stars mentioned in the caption are missing in the figure. Table 3. Could you include the meaning of A in the caption? In GISP2, should be A instead of r (third column)?

Telford, R. J., Li, C., & Kucera, M. (2013). Mismatch between the depth habitat of planktonic foraminifera and the calibration depth of SST transfer functions may bias reconstructions. Climate of the Past, 9(2), 859-870.

---

## Author Comment (AC2) · 28 Jan 2018

R. The paper by Jensen and co-authors is a very interesting contribution, very well written, which I find particularly useful for the paleoclimate community. The proxy surrogate re-construction method allows for a better understanding of the climate system beyond the spatial limitations (and variables to be reconstructed) by proxy data. I recommend the manuscript for publication, after a few corrections and some clarifications.

C. Thank you for your comments and review. Comments and some clarifications marked with C below.

[Figure]

R. Page 5. Data Pool. I assume the authors reconstruct SST at 10 m following methodology by Kucera et al. 2005 within the MARGO framework. However, Telford et al. 2014 demon- strated that this depth rarely is the most significant for fossil planktonic foraminifera assemblages in the North Atlantic, and more sensitive to subsurface conditions. This should be acknowledged by the authors. I may understand that for their purposes, this is not a very important issues, but it should be at least explained how this may affect the proxy-model comparison.

C. Thank you for the comment. We acknowledge the fact that the depth habitat of planktic foraminifera varies in the upper ocean and that this could lead to biases based on the chosen calibration depth. However, the problems with mismatches between depth habitat of planktic foraminifers and calibration depth of SST transfer function are primarily related to tropical areas. In the northern North Atlantic these problems are much less pronounced. Telford et al. (2013) showed that north of 25° N the reconstructions for different depth are very similar. We will add a piece of text that addresses this issue.

R. Page 6. L. 22. How the larger age uncertainty for cores 3,4 and 14 may affect the comparison between the surrogate and proxy time series? Could you add some sentence about this?

C. We will discuss this further and add some sentences. The results are not sensitive to dropping these 3 cores.

R. Page 10. L. 13. r instead of r2. Figure 7. Black stars mentioned in the caption are missing in the figure. Table 3. Could you include the meaning of A in the caption? In GISP2, should be A instead of r (third column)?

C. Thank you for these corrections, we will fix accordingly

Telford, R. J., Li, C., & Kucera, M. (2013). Mismatch between the depth habitat of planktonic foraminifera and the calibration depth of SST transfer functions may bias

reconstructions. Climate of the Past, 9(2), 859-870.